# Recursive splicing discovery using lariats in total RNA sequencing

Emma R Hoppe[1,2,3] , Dylan B Udy[1,2] , Robert K Bradley[1,2,3]

**Recursive splicing is a non-canonical splicing mechanism in which an intron is removed in segments via multiple splicing reactions. Relatively few recursive splice sites have been identified with high confidence in human introns, and more comprehensive analyses are needed to better characterize where recursive splicing happens and whether or not it has a regulatory function. In this study, we use an unbiased approach using intron lariats to search for recursive splice sites in constitutive introns and alternative exons in the human transcriptome. We find evidence for recursive splicing in a broader range of intron sizes than previously reported and detail a new location for recursive splicing at the distal ends of cassette exons. In addition, we identify evidence for the conservation of these recursive splice sites among higher vertebrates and the use of these sites to influence alternative exon exclusion. Together, our data demonstrate the prevalence of recursive splicing and its potential influence on gene expression through alternatively spliced isoforms.**

## Introduction

RNA splicing is the process by which introns are removed from precursor mRNAs to form functional mature mRNAs and long non-coding RNAs, a ubiquitous process among eukaryotes essential for proper gene expression (reviewed in Lee & Rio [2015]). Alternative splicing broadly refers to the phenomenon in which the use of different splice sites produces different mature mRNA sequences that often code for different protein isoforms (reviewed in Nilsen & Graveley [2010], Ule & Blencowe [2019], and Kelemen et al [2013]); the widespread usage of alternative splicing in the transcriptome greatly expands and diversifies the proteome without increasing the genome size to the same degree (Nilsen & Graveley, 2010; Blencowe 2017). The breadth of alternative splicing within a transcriptome correlates with the complexity of the organism (Kim et al, 2004, 2007), with >95% of human genes displaying some form of alternative splicing (Pan et al, 2008; Wang et al, 2008). Such

prevalence among higher eukaryotes implies that alternative splicing has an important contribution to the functional complexity of these organisms (Barbosa-Morais et al, 2012; Merkin et al, 2012).

Efficient removal of intronic sequences is essential for splicing; failure to do so leads to mature mRNAs that do not, for example, code for the correct protein sequence. Most splicing reactions are thought to proceed in a single set of two transesterification reactions (Padgett et al, 1984; Wilkinson et al, 2020), leading to the removal of the intron as a single lariat structure that is rapidly degraded (Padgett et al, 1984; Ruskin et al, 1984; Ruskin & Green, 1985). However, some introns are excised as multiple pieces across multiple reactions, a process referred to as recursive splicing (RS) (Hatton et al, 1998; Burnette et al, 2005; Kelly et al, 2015; Pulyakhina et al, 2015; Gazzoli et al, 2016; Conboy, 2021; Gehring & Roignant, 2021). Previous work characterizing recursive splicing has focused on long introns (>50 kb) (Hatton et al, 1998; Burnette et al, 2005; Sibley et al, 2015), with the rationale that there exists a maximum length for intron excision that preserves the accuracy of splicing (Burnette et al, 2005; Shepard et al, 2009; Suzuki et al, 2013; Duff et al, 2015; Sibley et al, 2015; Pai et al, 2018) and/or maintains proper coordination between the spliceosome and RNA polymerase (Pai et al, 2018; Zhang et al, 2018).

Empirically identifying recursive splice sites is inherently difficult because of the following: (1) the transient nature of intron-lariat structures (Ruskin & Green, 1985; Mohanta & Chakrabarti, 2021), (2) the dearth of RNA-seq reads derived from lariats because of poly(A) selection in most sequencing library preparations, and (3) the low probability of lariat-derived sequencing reads traversing the recursive splice site owing to the large size of many introns. In addition, the frequency at which recursive splice sites are chosen over conventional sites remains unclear. Site preference may be stochastic, depend on intron identity, or be regulated by as-yet-unknown mechanisms (Radtke et al, 2017 *Preprint*; Wan et al, 2021). The rarity of recursive splicing relative to conventional splicing may also present challenges. Nevertheless, previous work has used novel approaches to discover recursive splice sites. Foundational minigene experiments in *Drosophila* demonstrated that the *Ubx* cassette exon can be excluded via the usage of the cassette exon's 5'-terminus as a recursive splice site, which excises

[1]Computational Biology Program, Public Health Sciences Division, Fred Hutchinson Cancer Center, Seattle, WA, USA   [2]Basic Sciences Division, Fred Hutchinson Cancer Center, Seattle, WA, USA   [3]Department of Genome Sciences, University of Washington, Seattle, WA, USA

Correspondence: rbradley@fredhutch.org

the long stretch of the intron–exon–intron sequence in multiple pieces (Hatton et al, 1998). Work from the same group used splice site sequence preferences to search for potential recursive splice sites among all annotated introns in the *Drosophila* genome (Burnette et al, 2005); this analysis yielded 165 potential recursive splice sites, primarily in long introns >10 kb, and a subset were experimentally verified.

The increased prevalence of RNA-seq has led to more high-throughput analyses for identifying potential recursive splice sites from sequencing data. Two groups were searched for novel recursive splice sites in *Drosophila* and humans (Duff et al, 2015; Sibley et al, 2015) using total RNA-seq data to identify "sawtooth" sequence patterns—named for the characteristic coverage patterns in intronic regions because of reads from splicing intermediates: peaks at 5′ splice sites followed by approximately linear decreases in signal intensity approaching 3′ splice sites. Sibley et al (2015) identified recursive splice sites in human genes for the first time, although restricting the analysis to only genes with introns >150 kb meant the search was not exhaustive. Subsequent studies used 4-thiouridine labeling to sequence RNA shortly after transcription to enrich for nascent RNA and more readily identify recursive splice sites in *Drosophila* (Pai et al, 2018) and human cells (Zhang et al, 2018). Importantly, the recursive splice site identification in human cells was restricted to introns >5 kb (Zhang et al, 2018), limiting the total number of introns analyzed.

The regulatory capacity of recursive splicing and its ability to modulate gene expression have not been characterized in detail. To assess potential mechanisms, the identification of recursive splice sites in a broader range of introns is necessary. Therefore, we sought to build upon the work of previous studies by searching for novel recursive splice sites in constitutive introns of all sizes in the human transcriptome using an unbiased approach to analyze previous sequencing datasets potentially enriched for lariat-spanning reads and employing stringent filters to identify the highest confidence recursive splice sites.

# Results

## Global annotation of recursive spicing using total RNA sequencing

The rarity of lariat-spanning reads, specifically from recursive splicing, in traditional RNA-seq datasets makes thorough annotation of such sites difficult. To maximize the likelihood of identifying lariats associated with recursive splicing, we chose to analyze datasets without poly(A) selection (Table S1), a preparation method that depletes lariats (see the Materials and Methods section for more details).

Our method for recursive splice site identification differed from those of other groups that restricted their searches to long introns (Sibley et al, 2015; Zhang et al, 2018). We searched all constitutive introns in the human genome for $Y_{5+}N_{0-4}YAGGT$ motifs, representing a minimal 3′ splice site (short polypyrimidine tract and 3′ splice site YAG) joined to a minimal 5′ splice site (GT), to identify potential recursive splice sites. For each site, we chose five "decoy" sites at random positions in the same intron and filtered out those likely to be involved in cryptic splicing. Decoy sites are important for

calculating the empirical false discovery rate (FDR) in recursive splice site identification. Each putative and decoy recursive splice site was paired with the annotated 5′ and 3′ splice sites in the respective intron and with other putative RS or decoy sites within the same intron, creating pairs of junctions for aligning with reads.

For empirically identifying recursive splice sites from the putative sites, we aligned RNA-seq reads to our junctions using a "split-read" approach described previously (Mercer et al, 2015; Pineda & Bradley, 2018). Briefly, the first 20 nucleotides of each junction, corresponding to the 5′ splice site region, were mapped to prefiltered reads (Fig 1A). Successfully aligned reads were trimmed to remove the 5′ splice site region and mapped to the 3′ splice site region (last 250 nucleotides of the junction; Fig 1A). Reads that aligned successfully to the recursive splice site junctions were further filtered as described later (Fig S1A). Examples of recursive splice junctions supported by sequencing reads are shown in Fig 1B, including reads that contain the 5′ portion of the recursive splice site (green) and reads in which the recursive splice site is inferred downstream from the branchpoint position found in the read (blue). Additional details on putative recursive splice site determination and split-read mapping are available in the Materials and Methods section.

## Lariat sequencing achieves low empirical FDR with key filters

The pipeline for empirically identifying recursive splice sites yielded a substantial number of such sites. However, the rate of false-negative detection, based on the number of decoy sites that were called as recursive splice sites, was higher than anticipated (Fig S1A), indicating that the traditional sequence features used to identify lariat reads are insufficient for identifying genuine recursive splice sites with a low FDR. Therefore, we employed a series of filtering steps to minimize the number of false negatives while still identifying novel recursive splice sites with much higher confidence (Fig S1A).

The first filtering step required at least one read with a mismatch at the branchpoint site; such a mismatch is created from the low fidelity of reverse transcriptase when traversing the 2′–5′ phosphodiester linkage at the branchpoint nucleotide (Vogel et al, 1997; Gao et al, 2008) and is indicative of bona fide lariat sequences. The next filtering step selected only "high-confidence" recursive splice junctions that mapped to reads as described in Pineda and Bradley (2018). Briefly, this required (1) >5% of reads mapping to a specific junction to contain a mismatch at the branchpoint nucleotide and no other mismatches in the 3′ splice site portion of the read and (2) the sequences upstream and downstream of the branchpoint be unique to the intron of interest and not found elsewhere in the genome (to abrogate the misannotation of gene duplications as recursive splice sites). This filtering step had the largest impact on reducing the empirical FDR (Fig S1A, column 3).

An additional filter excluded mapped recursive splice sites within 100 nucleotides of annotated splice sites (Fig S1A, column 4) as a heuristic to exclude cryptic alternative 5′ and 3′ splice sites. In addition, mapped sites were removed if the 20 nucleotides required for mapping the recursive site (downstream of the 5′ splice site or upstream of the branchpoint) overlapped with a repeat or low-complexity region (Fig S1A, column 5) to reduce uncertainty in the

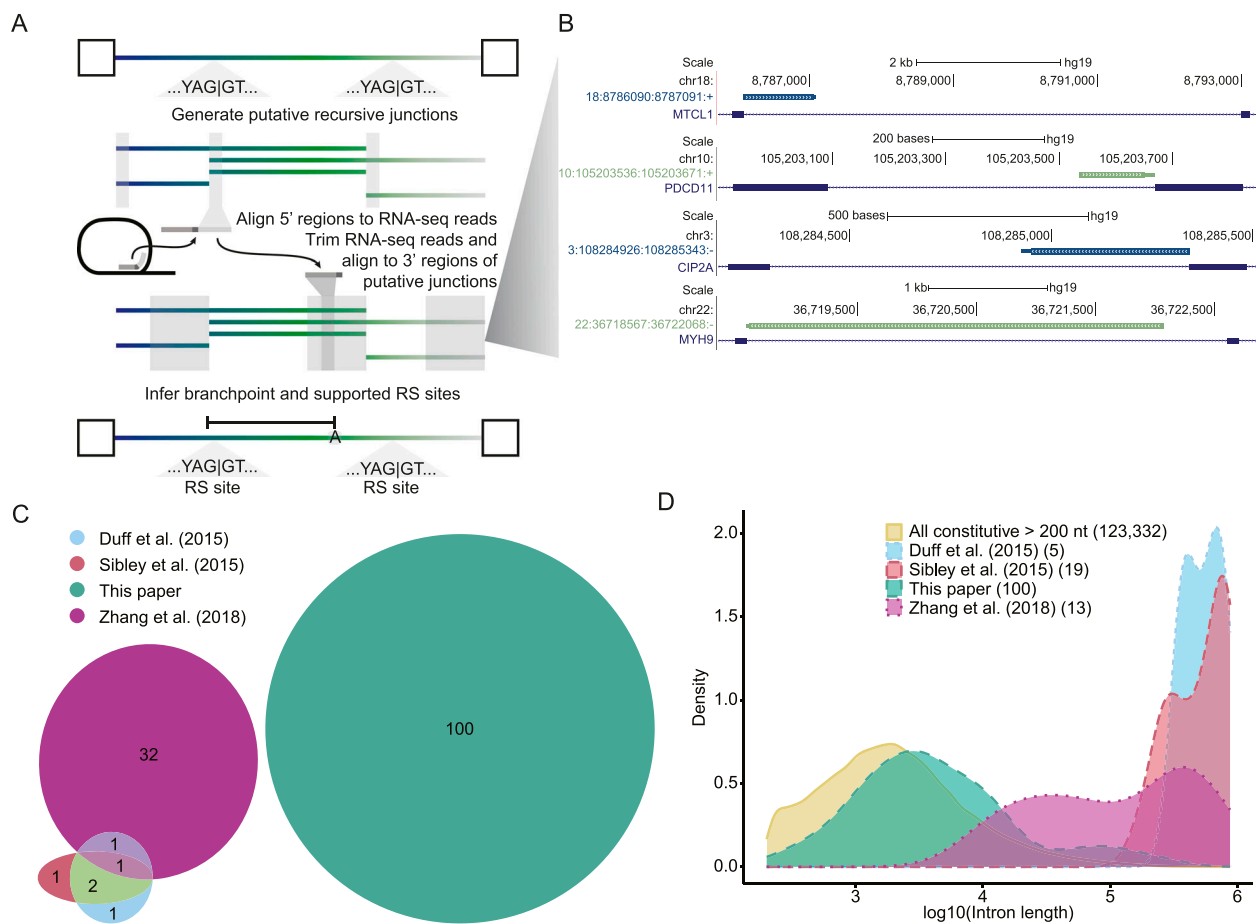

**Figure 1. Global annotation of recursive splicing using lariats in total RNA sequencing.**
**(A)** Overview of the recursive lariat mapping algorithm. Intronic sites matching the sequence YAGGT with an upstream polypyrimidine tract containing at least five adjacent pyrimidines separated by a span of 0–4 N (see the Materials and Methods section "Lariat detection" for sequence motif) were identified as putative recursive sites. These sites were mapped against reads from total RNA-seq, after removing genome- and transcriptome-mapping reads, in a stepwise, split fashion to ensure the inverted read expected of lariats and enable us to identify branchpoint mismatches. **(B)** Representative recursive lariats. Thick bar, the region between the mapped 5′ splice site and the branchpoint (loop of the lariat); adjacent thin bar, the inferred branchpoint to 3′ splice site; navy, conventional 5′ss to RS 3′; and green, RS 5′ss to conventional 3′ss. The plot is derived from the custom track generated from the University of California at Santa Cruz Genome Browser (Meyer et al, 2013). **(C)** Overlap of human RS sites identified as high confidence in previous studies, predominantly using sawtooth mapping, and those in the current work. **(D)** Length distributions for introns in which recursive sites were identified across different studies compared with constitutive introns. Counts of supported junctions with intron length annotations in parentheses.

mapping. Previous work has shown that the branchpoint to the 3′ splice site is predominantly constrained within a range of 10–60 nucleotides (Taggart et al, 2012, 2017; Mercer et al, 2015; Pineda & Bradley, 2018). Only mapped recursive splice sites in which the distance between the branchpoint and the 3′ splice site fell within this range were included for further analysis (Fig S1A, column 6).

A substantial portion of remaining reads that mapped to decoy random-site junctions were best explained as originating from self-primed amplification—whereby linear reads circularized or took on the appearance of being circularized by ligating up/downstream with themselves or a daughter strand during the PCR amplification step of RNA-seq library preparation—and did not represent reads from bona fide recursive splice sites based on manual inspection. These types of rare potential technical artifacts were removed in the last filtering step by comparing the five nucleotides upstream of the 5′ splice site with the four nucleotides upstream of the

branchpoint nucleotide plus the branchpoint and excluding reads with exact matches (Fig S1A). In aggregate, these filters produced an overall empirical false discovery rate of 2.51% across all of our datasets, with a range of 0–3.9% for the five individual datasets included (Fig S1B; Table S2).

## Comparison of high-confidence recursive splice sites with previously identified sites

The methodological differences in our approach, compared with previous studies (Duff et al, 2015; Sibley et al, 2015; Zhang et al, 2018; Wan et al, 2021), led to the identification of previously unknown recursive splice sites. All 100 sites we identified were absent from previously annotated high-confidence sets of recursive splice sites (Fig 1C), whereas the previous sets had at least one recursive splice site in common with each of the other

previous sets (Fig 1C). However, our sites did have limited overlap with the lower confidence method of split-read mapping from Sibley et al (2015), including one site that met their more restrictive recursive motif (Fig S2) and nine additional sites (data not shown). Together, this indicates that we have yet to reach saturation of the human recursive splicing annotation. The limited overlap that we observed could also arise in part from differences in the cell types that were analyzed in each study. Our analysis used a diverse selection of cell lines and patient-derived fibroblasts (Table S1), which were distinct from the cell types analyzed in other works (Duff et al [2015]: selected adult tissue samples; Sibley et al [2015]: adult brain tissue; Zhang et al [2018]: ovarian, embryonic stem cell, and embryonic stem cell–derived forebrain cell lines; and Wan et al [2021]: bronchial epithelial cell line). In addition, the length of the introns from which the recursive splice sites were identified in previous work is much larger than that of our work (Table 1 and Fig 1D), reflecting the specific selection for large introns in previous analyses (Sibley et al, 2015; Zhang et al, 2018). Our study demonstrates that smaller introns, too, which were largely excluded by previous efforts, are capable of recursive splicing. These comparisons highlight the value of unbiased analysis in all constitutive introns to discover novel recursive splice sites, and the need for the continuation of work on this topic, as it remains likely that many sites remain undiscovered.

## Lariat sequencing identifies recursive splicing in diverse introns

With the identification of these high-confidence recursive splice sites, we sought to investigate the nature of the introns from which they were derived. For every constitutive intron in the human genome, we modeled the likelihood of an intron yielding an informative lariat read as a function of the intron length and estimated the proportion of reads we would expect to be informative for identifying recursive splice sites (see the Materials and Methods section for additional modeling details). As introns increase in length, the fraction of informative reads decreases (Fig 2A), as expected because longer introns contain more sequences not immediately adjacent to the branchpoint nucleotide (and thus indistinguishable from an intronic sequence originating from genomic DNA or unspliced intermediate RNA reads). Our modeling also indicates that longer read lengths lead to a higher likelihood of an informative read for any given intron length (Fig 2A).

We compared the expected intron length distribution with the length distribution of the identified recursive splice site introns (Fig 2B). Using the proportion of informative lariat read calculations from Fig 2A, we calculated the probability-weighted distribution of expected intron lengths (Fig 2B, gray curve) and the unweighted distribution (Fig 2B, orange curve) for all constitutive introns. Interestingly, the sizes of the identified recursive splice site introns displayed a bimodal distribution, with peaks mimicking the peaks in

**Table 1. Characteristics of recursive splicing identified in this work compared with previous studies.**

| Approach | | Sites with motif | | | Intron length (kb) |
|---|---|---|---|---|---|
| | n | AGGT | YAGGT | +polyY tract[a] | Median (min, max) |
| Duff et al (2015) | | | | | |
| Sawtooth | 5 | 5 (100%) | 5 (100%) | 4 (80%) | 552.3 (370.4, 766.9) |
| Sibley et al (2015) | | | | | |
| Sawtooth | 19 | 9 (47%) | 9 (47%) | 7 (37%) | 547.4 (171.8, 874.9) |
| Split-read (>150-kb introns) | 2,520 | 849 (34%) | 469 (19%) | 115 (5%) | 195.7 (15.3, 1,097.9)[b] |
| Split-read (1- to 150-kb introns) | 57,343 | 17,589 (31%) | 10,219 (18%) | 2,091 (4%) | n.p. |
| Zhang et al (2018) | | | | | |
| Lariats | 21 | 21 (100%) | 20 (95%) | 9 (43%) | n.p. |
| Lariats + sawtooth | 3 | 3 (100%) | 3 (100%) | 2 (67%) | 370.4 (32.9, 552.3) |
| Sawtooth | 10 | 10 (100%) | 10 (100%) | 4 (40%) | 100.8 (11.8, 552.3) |
| Split-read only | 330 | 330 (100%) | 280 (85%) | 114 (35%) | 36.4 (5.1, 1,055.3) |
| Wan et al (2021) | | | | | |
| Lariats | 5,983 | 1,589 (27%) | 949 (16%) | 157 (3%) | n.p. |
| This study | | | | | |
| Lariats (additional filtering) | 100 | 100 (100%) | 100 (100%) | 100 (100%) | 3.4 (0.2, 247.9) |

[a]motif = $Y_{5+}N_{0-4}YAGGT$.
[b]n = 483 with annotated introns.
Sites here include all those provided that meet the filtering described in the Materials and Methods section of the respective articles for the primary analysis, typically not including the motif filtering that may be used elsewhere in the article (excluding Duff). Intron annotations were used from only the originating article, but from whichever class of intron they could be obtained. Both site numbers and intron lengths were weighted to accommodate multiple intron/gene annotations if they arose. n.p., annotation not provided by authors.

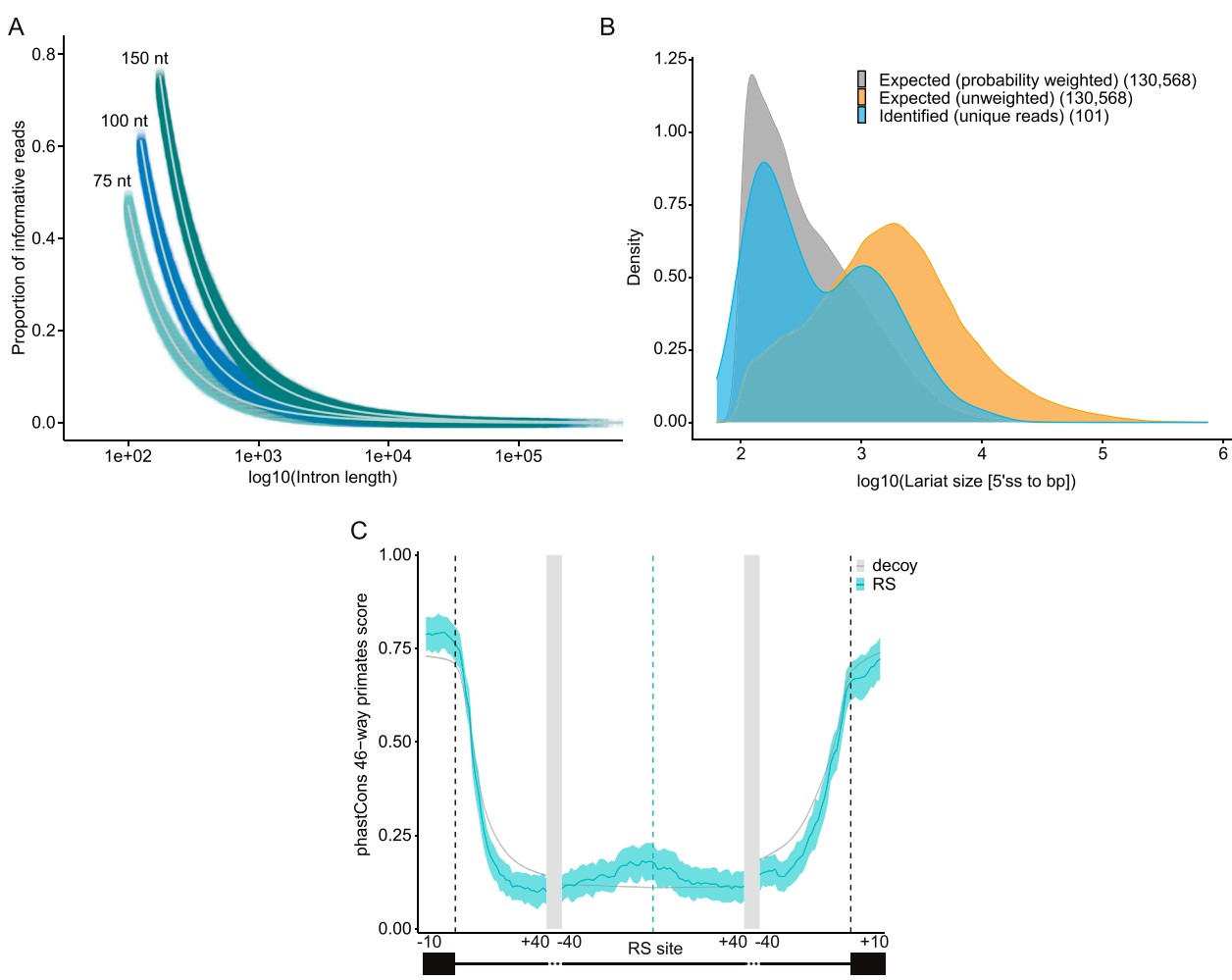

**Figure 2. Lariats follow expected size distributions, and RS sites show conservation enrichment.**
**(A)** Estimated probabilities of a read of a given length being informative for lariat detection given the mapping constraints used in this work (see the Materials and Methods section "Modeling informative reads" for parameters of informative reads). 75, 100, and 150 nt indicate the read length parameter used for each set of simulations. **(B)** Comparison of expected lariat sizes with our mapped lariats. The distributions of lariat sizes from constitutive introns, assuming equal expression, with and without weighting according to the probability of identifying the lariat (for 100-nt reads) compared with our recursive lariat reads. **(C)** PhastCons scores for the regions around the RS site and flanking exons for primates. RS sites identified in this analysis, blue; randomly chosen decoy sites, gray. Light blue and light gray areas indicate 95% confidence intervals from RS and random decoy sites, respectively, calculated using bootstrapping (replicates = 100).

the weighted distribution and the unweighted distribution (Fig 2B, blue curve). Because our method preferentially identifies recursive splice sites from shorter introns, the first peak is expected, but the second peak indicates that recursive splicing may indeed be enriched to an extent in larger introns, where the focus of study has centered to date. It is worth noting that we did not control here for gene expression, which is likely to impact lariat discovery. Furthermore, our annotation to date of recursive sites appears not to be fully saturated so the true distribution of intron lengths implicated in recursive splicing remains to be uncovered. Nonetheless, these data indicate that the recursive splicing exists over a range of intron lengths.

To assess the level of evolutionary conservation at these sites, we analyzed the phastCons scores of the upstream and downstream regions of the site and surrounding conventional exons. The mean base-level conservation among primates was increased

modestly immediately upstream of RS sites compared with randomly chosen decoy sites (Fig 2C; bootstrap support >95%). We also compared the level of conservation for placental mammals, which was slightly increased in the RS site region (Fig S3A; bootstrap support >95%), and among vertebrates, which was not increased (Fig S3B). Given the difficulty of alignment in intronic regions, a low level of conservation still may indicate selection at these sites. It also may point to the variability of conservation at such sites, in accordance with the finding that recursive splicing appears to be stochastic at some sites and seemingly regulated at others (Wan et al, 2021).

Finally, we compared the relative rates of identification of conventional and recursive lariats for the introns in which we identified recursive splicing to determine whether recursive splicing appeared to be occurring infrequently or stochastically, as suggested by Wan et al (2021), versus as a more common or

dominant mode of splicing. Approximately half of the introns (46 of 100) with recursive lariats had at least one conventional lariat that we identified. For short introns (<1,000 nt), we observed a preference for lariats arising from conventional versus recursive splicing (Fig S4A and B). A Spearman rank correlation performed on the relative number of unique recursive lariats with mismatches compared to the intron length in those introns with at least one lariat of each type revealed a positive association between intron length and the relative number of recursive lariats ($\rho$ = 0.4912, $P$ = 0.0005272). These data suggest that certain introns identified here, especially smaller introns, may be primarily subject to recursive splicing as a stochastic and less frequent mode of splicing, whereas others show a preference for recursive over conventional splicing.

### Distal exonic RS sites contribute to exon exclusion

Blazquez et al (2018) described RS-exons where after the upstream intron is spliced out, the reconstituted 5′ splice site in the exon is used for the removal of both the exon—here called proximal recursive exons—and the downstream intron. Our results here demonstrate there is likewise exon exclusion using a distal exonic 3′ splice site. The choice of this distal recursive splice site enables the upstream intron and exon to be removed in one splice reaction, followed by the splicing of the conventional downstream intron (Fig 3A).

We initially observed potential support for this mechanism in sequence logo plots of cassette exons. The final dinucleotide of exons, both cassette and constitutive, is commonly AG (Fig 3B; 56.2% cassette and 55.6% constitutive), forming a minimal recursive splice motif. We hypothesized that a subset of these sites had sufficient pyrimidines upstream to be recognized as a 3′ splice site to enable the use of this distal exonic site. We calculated the maximum entropy (MaxEntScan) scores (Yeo & Burge, 2004) to assess how closely they adhered to the sequence of constitutive introns; 10.2% of cassette exons maintain a MaxEntScan score of at least 4.07 (5th percentile of constitutive 3′ splice sites), suggesting the potential for recognition as a 3′ splice site, with 1.62% of cassette exons (780) having a MaxEntScan score at least as high as the 50th percentile of constitutive 3′ splice sites. Given the ability of AG-dependent introns to be spliced with very short or weak polyY tracts as long as they maintained a strong YAG 3′ splice site (reviewed in Moore [2000]) and the potential constraints on polyY tract length in coding exons, we moved forward looking at sites with a YAG as the last trinucleotide of the exon.

To investigate whether there was evidence for the use of such sites as distal recursive exons, we analyzed the rate of inclusion of cassette exons that ended in this minimal YAG compared with all others, including those that ended in AG. At the 50th percentile for inclusion across the median isoform inclusion rate of the 16 BodyMap tissues, exons that ended in YAG were 23% less than those that ended in another trinucleotide (Fig 3C). The overall distribution of the median p.s.i. values of YAG-ending cassette exons was likewise significantly left-shifted (Kolmogorov–Smirnov test, $P$ = $6.51 \times 10^{-37}$). These data suggest that the presence of YAG in the 3′ splice site of a cassette exon is a strong signal for its exclusion.

Given the evidence for the use of distal recursive exons, we performed a modified version of the intronic recursive splicing analysis above. We determined that the critical filter for distal

recursive exons was to require that the branchpoint fall within the body of the exons. This enabled the exclusion of alternative 3′ splice sites or longer than typical branchpoint to 3′ splice site lengths. In total, we found evidence of 10 distal recursive exons (Table S3), which each passed a manual review for sequence and mapping irregularities that had been filtered for in the larger intronic recursive site set. A selection of mapped lariats is shown in Fig 3D. The MaxEnt scores for these lariat-supported distal exonic RS sites ranged from −2.62 to 11.61 with a median of 7.34 (Table S3), indicating that they largely, but not exclusively, adhere well to the U2 consensus 3′ss sequence. Some sites are supported by multiple datasets and showed a diversity of branchpoints; for instance, the 9th exon in CAPN10 (hg19 2:241536098:241536359:+) has four unique branchpoints, one of which is supported by lariats from two datasets (Table S3).

In further support of the usage of distal exonic RS sites, two of the sites identified in this analysis are located at the junction between two mutually exclusive adjacent exons (Fig 3D), known as a dual specificity site (Zhang et al, 2007). The maintenance of dual specificity sites in the genome demonstrates the existence and the usage of distal exonic 3′ splice sites, and suggests that they also may rely upon exon definition of a downstream exon. Although relatively rare in the genome, dual specificity sites demonstrate that both the distal and proximal exonic recursive splice sites may be used alternatively to include or exclude adjacent exons. What genomic signals are associated with the use of dual specificity sites and exonic recursive sites more broadly remains unclear, but they present an intriguing possibility as an evolutionary intermediate for exon birth and death.

To further investigate whether distal exonic RS sites may be involved in exon birth or death, we compared the proportions of internal exons that ended in YAG in internal exons across age categories of exons using the annotations from Corvelo and Eyras (2008). Corvelo and Eyras (2008) included only those exons that were protein-coding and without an alternative 3′ or 5′ splice annotation. Exons annotated as primate-specific and cassette exons that were not included in an age category had significantly higher proportions that ended in YAG than unassigned constitutive exons and vertebrates and older exons (Fig S5; binomial exact test, primate-specific: $P$ = 0.0079; cassette: $P$ = $1.7 \times 10^{-5}$). Likewise, primate- and mammalian-specific exons were found to have a significantly higher proportion ending in YAG than vertebrate and older exons (primate: $P$ = 6.39 × $10^{-4}$; mammalian: $P$ = 8.82 × $10^{-3}$). Although the comparisons are limited by the relatively small number of age-categorized exons, there are clear enrichment for YAG at the end of the most recently evolved exons, as well as those that are alternatively included, and apparent depletion from the oldest maintained exons. This enrichment for YAG in evolutionarily young protein-coding exons suggests that distal RS-exons may indeed play a role in the birth and perhaps the death of exons, and bears further exploration.

## Discussion

Since its initial discovery in *Drosophila*, much of the work on recursive splicing has focused on its prevalence in long introns. Here, we present evidence that recursive splicing occurs more

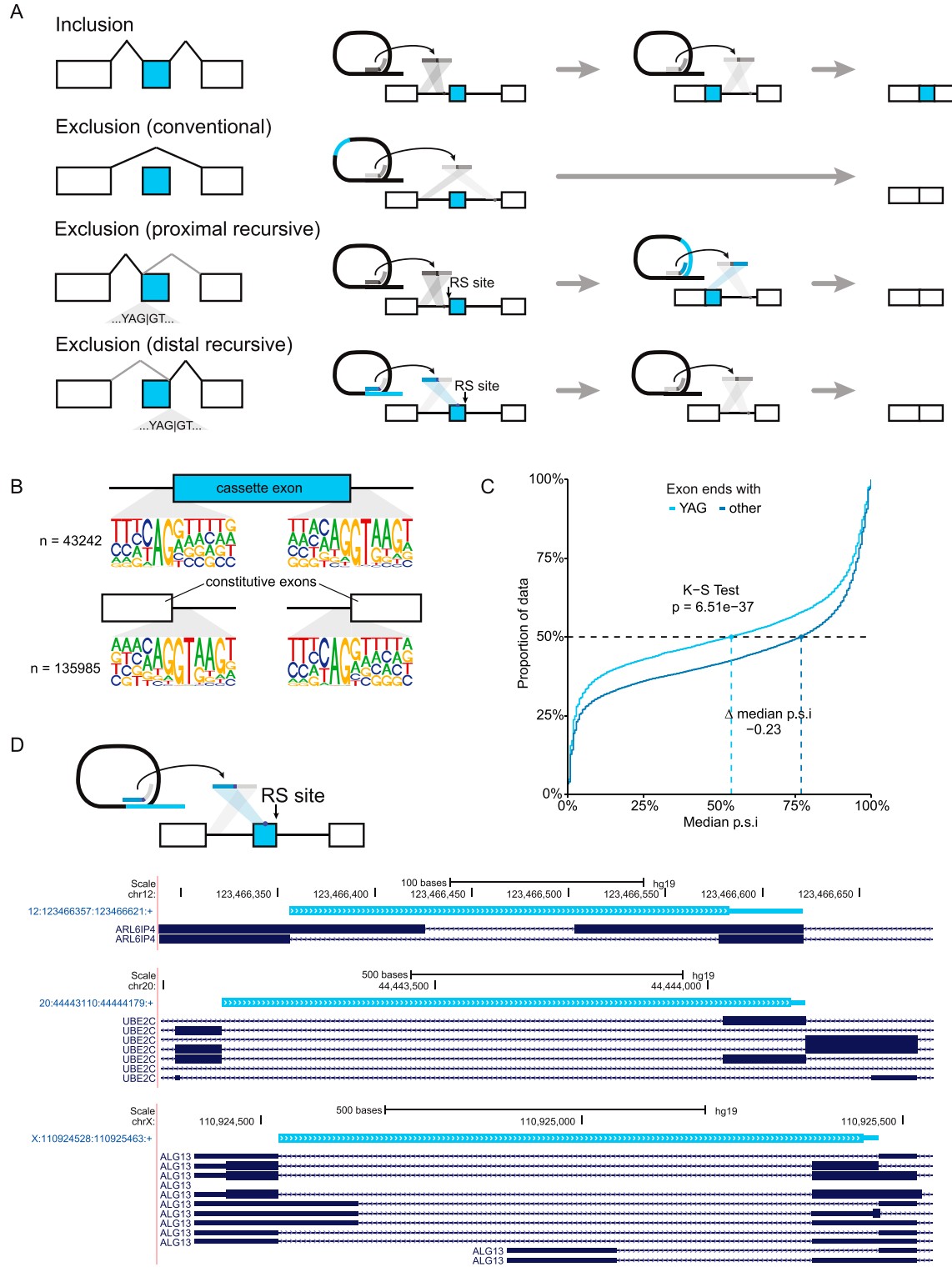

**Figure 3. Distal exonic recursive splicing may contribute to exon exclusion.**
**(A)** Schematic of exon inclusion or exclusion using conventional or exonic recursive splice sites. **(B)** Sequence logo plots of cassette exons aligned with constitutive exons showing strong enrichment for AG|GT at the distal end of exons. **(C)** Proportion of data with median percent spliced in (p.s.i.) values across BodyMap tissues of cassette exons ending in YAG or any other trinucleotide. K–S test, Kolmogorov–Smirnov test. **(D)** Representative distal exonic recursive lariats. Thick bar, the region between the mapped 5′ splice site and the branchpoint (loop of the lariat); adjacent thin bar, the inferred branchpoint to 3′ splice site. The plot is derived from a custom track from the University of California at Santa Cruz Genome Browser (Meyer et al, 2013) and contains all unique, annotated isoforms within the given window.

broadly—both in small- to medium-length introns and at the distal end of exons—by employing an unbiased lariat sequencing approach to identify recursive splicing genome-wide.

For identifying RS sites from lariat reads, we chose a sequence motif that adhered to the U2 consensus sequence, which seemed likely to capture a substantial number of sites, both U2-type and U12-type, with minimal computational requirements. This analysis yielded 100 RS sites that strongly adhered to the U2 3′/5′ consensus sequence in constitutive introns. We limited the analysis to constitutive introns, so there would be somewhat less uncertainty about the origins of the lariat, although alternative introns are a potentially rich source of recursive splicing and would be of vital interest for future work. Importantly, our method assessed the false discovery rate of lariat sequencing and found it highly dependent on the sequencing preparation method used. Although poly(A)-selected RNA-seq can contribute meaningfully to conventional lariat counts (Pineda & Bradley, 2018), it was a poor fit for recursive analyses given the high FDR (data not shown). Datasets with more specific selection—such as those that use RNase R to enrich for non-linear RNA—yielded the lowest FDRs, and poly(A)-minus datasets also produced reasonable results. The filtering steps employed to reduce the FDR identified sequencing artifacts that are called as recursive splice sites, including apparent self-priming amplification that caused circularization of DNA using at least five bases. Unexpected sequencing artifacts such as these justify the use of stringent filters for these analyses and emphasize the importance of calculating empirical FDR.

The filtering steps used to minimize FDR in our analyses revealed multiple aspects of sequencing that affect data quality. First, as expected from our modeling, increased sequencing read lengths lead to a higher proportion of informative reads and extend the size of lariats we could practically identify. As long-read sequencing depth increases and short-read sequencing length increases, our ability to identify additional recursive sites in introns in the 10–100 kb size range will improve. Second, some selection is required for optimal sequencing of lariats. Although removing poly(A) sequences (in addition to standard rRNA depletion) appears sufficient for achieving a low FDR, enriching further with RNase R treatment or enriching for specific introns/lariats with a pull-down would be ideal for targeted RS lariat sequencing experiments. Third, the selection of small RNAs before reverse transcription appears to eliminate the bulk of lariats. The statistical size constraints described previously limit the usefulness of these datasets, so we do not recommend the use of small RNA datasets for RS site identification. Fourth, patient samples that otherwise met the above criteria are not useful with this method, regardless of tissue of origin. We hypothesize that this is due to sample processing and handling protocols that may lead to the degradation of lariats, as has been observed with other classes of RNAs (Dvinge et al, 2014).

Although recursive splicing has been proposed to be important for *Drosophila* developmental timing (Duff et al, 2015; Pai et al, 2018), splicing fidelity (Burnette et al, 2005; Shepard et al, 2009; Duff et al, 2015; Sibley et al, 2015; Pai et al, 2018), and timing between RNA polymerase and the spliceosome (Pai et al, 2018; Zhang et al, 2018) in long introns (>10 kb), it is unclear what purpose recursive splicing

serves in shorter human introns. Recent work that tested recursive splice sites using a reporter found that some sites were necessary for proper splicing of the intron (Radtke et al, 2017 *Preprint*), which, combined with the conservation of recursive splice sites among primates (Fig 2C) in our data, implies that recursive splicing is more than just a stochastic process in at least a subset of introns. Previous studies using single-molecule imaging (Wan et al, 2021) and 4-thiouridine labeling (Zhang et al, 2018) show that both RS and conventional splicing can occur in the same intron, and the use of either splicing mechanism is often cell type–specific (Zhang et al, 2018). Our work here indicates that the frequency of use of recursive versus conventional splicing may be a function of intron length, with stochasticity playing a stronger role in smaller introns. However, further work will be needed to confirm this observation and elucidate the role, if any, recursive splicing plays in regulating alternative introns, as the work here primarily focuses on constitutive introns.

A provocative hypothesis, supported by our finding that the motif suggestive of distal recursive exons is enriched among evolutionarily young exons, is that RS sites contribute to exon birth and death. Such sites could provide variation in splicing—without substantial fitness costs—until they are paired and selected as distal/proximal RS sites flanking a new exon or eliminated. Our analysis here identified lariats supporting the recursive usage of the 3′ splice site of dual specificity sites/distal exonic recursive splice site within coding exons (Fig 3D). However, we did observe a bias toward the support of distal recursive exons in UTRs (data not shown), in which case the protein-coding sequence would be unchanged. Zhang et al reported observing multiple protein products from genes with dual specificity sites (Zhang et al, 2007); the different proteins corresponded to mRNA isoforms differentially spliced using the dual specificity sites, providing evidence for the presence of functional 3′ splice sites within the coding sequence. Together with the work on proximal recursive exons (Blazquez et al, 2018), these data indicate that splicing happens within exonic sequences to some degree. Whether the same mechanisms of exon junction complex removal that promote the usage of proximal recursive splice sites (Blazquez et al, 2018) likewise promote the usage of distal recursive exons remains to be determined.

## Materials and Methods

### Genome annotations and alternative splicing identification

All analyses were performed with the hg19 (GRCh37) genome assembly. The transcriptome/splicing annotation used throughout was created by merging the Ensembl v.71.1 gene annotation (Flicek et al, 2013), the UCSC knownGene gene annotation (Meyer et al, 2013), and the MISO v.2.0 isoform annotation (Katz et al, 2010).

### Dataset selection and annotation

A manual search of the literature and datasets available in the Sequence Read Archive (SRA) was conducted to identify nascent

human RNA-seq datasets lacking poly(A) selection and, ideally, enrichment for circular RNAs or lariats via poly(A) depletion, RNase R, or inborn mutations in proteins involved in lariat degradation. Chosen runs were downloaded using the SRA toolkit and converted to FASTQ files (https://trace.ncbi.nlm.nih.gov/Traces/sra/sra.cgi?view=software). Files that failed to download three times were excluded. Annotation files were obtained via the SRA. Read counts were parsed from the SRA website for each SRR file using a custom Python script using the Pandas (McKinney, 2010), time, and Selenium libraries. A detailed list of the characteristics of the five datasets analyzed, totaling 117 samples and $1.22 \times 10^{10}$ reads, including sample characteristics, is included in Table S1.

## Lariat detection

### Generate putative and decoy junctions

Constitutive introns, defined as those contained in all RefSeq transcripts of each parent gene, were searched for potential recursive splice sites using the U2 consensus 3′ splice site motif appended with the minimal 5′ splice site: $Y_{5+}N_{0-4}YAGGT$. For each potential recursive splice site within each intron, five decoy sites were selected at random positions in the intron. These decoy sites were then filtered to remove those with an adjacent GT and those within the mapping range (250 nt) of a potential cryptic 3′ splice site (identified with the motif $Y_{5+}N_{0-4}YAG$), another potential recursive splice site, or an annotated 3′ splice site. In total, this method produced 202,395 unique putative recursive splice sites and 121,154 decoy splice sites. Each set of sites was used, independently, in conjunction with the annotated 5′ and 3′ splice sites to produce all pairwise junctions of potential 5′ and 3′ splice sites.

### Prefilter reads against genome and transcriptome

As described in Pineda and Bradley (2018), reads with fewer than 5% ambiguous bases were sequentially mapped against the transcriptome and then the genome (hg19/GRCh37) using Bowtie2. The following parameters were used with each mapping step: bowtie2 -× - -end-to-end –sensitive - -score-min L,0,-0.24 -k 1 - -n-ceil L,0,0.05 -U. Successfully aligned reads were discarded for subsequent steps.

### Map reads to 5′ and 3′ regions of identified junctions

The prefiltered reads were mapped to the putative recursive splice and decoy junctions described above, as detailed in Pineda and Bradley (2018). Briefly, the first 20 nt of each junction (conventional, putative RS, and decoy) was mapped to a Bowtie index created from the prefiltered reads. For successful 5′ alignments with no mismatches or indels where the reads aligned to a single 5′ splice site region, the original read was trimmed from the first nucleotide of the 5′ splice site alignment to the end of the read. These trimmed reads were mapped against a Bowtie index generated from the last 250 nt of each junction.

- 5′ Mapping: bowtie2 -× –end-to-end –sensitive –k 10000 –no-unal -f -U <FASTA file of 5′ splice site sequences>
- 3′ Mapping: bowtie2 -× <index file for 3′ splice sites> –end-to-end –sensitive -k 10 –no-unal -f -U

### Infer branchpoint positions and recursive sites from split-read alignments

Using the same methods described in Pineda and Bradley (2018), alignments were restricted to the best-scoring alignment for each read, those with inverted alignments indicative of lariat reads, those that mapped to sites within a single gene, and those with a single mismatch at the branchpoint position. The branchpoint position was defined as the last nucleotide of the trimmed read alignment.

Recursive splice sites were deemed to be supported if they had at least one read that supported their use as either a 5′ or 3′ splice site that fit the following filtering criteria. In cases where multiple RS sites acting as a 3′ splice site were supported by a single 5′-branchpoint mapping, a weight was assigned equal to 1/(number of possible RS sites); that is, if two nearby RS sites could have produced the 5′-bp mapping, each was assigned a weight of 1/2 in counts.

## Filter hits

### Criteria for high-confidence lariat detection

Mapped junctions were required to meet the criteria for high confidence outlined in Pineda and Bradley (2018): one or more reads with a mismatch at the branchpoint but no additional mismatches or indels in the 3′ splice site region of the read; ≥5% of mapping reads have a mismatch at the branchpoint but no additional mismatches or indels in the 3′ splice site region of the read; and unique sequences at the 5′ splice site and the region upstream of the branchpoint.

### Self-primed

To filter out reads that originated from self-primed amplifications during RNA-seq processing, the 5 nt of sequence upstream of and including the identified branchpoint was compared with the 5 nt upstream of the identified 5′ splice site, taking into account the strandedness of the sequence and read. Sequences with exact matches were discarded.

### Low-complexity regions and simple repeats

The RepeatMasker .out file for hg19 was obtained from UCSC (https://hgdownload-test.gi.ucsc.edu/goldenPath/hg19/bigZips/hg19.fa.out.gz). If the 20-nt region (minimum mapped region) including and downstream of the 5′ splice site or upstream of and including the branchpoint overlapped a region annotated by RepeatMasker as a simple repeat or a low-complexity region, it was discarded.

### Proximity to previously annotated splice sites and annotated exons

Mapped junctions were excluded if the putative recursive splice site fell within the 100 nt of an annotated splice site of the same type (5′ or 3′). This was performed by first using the resize function from GRanges (Bioconductor, version 3.14) fixed to either "start" (5′ splice site regions) or "end" (3′ splice site regions) on a list of all intronic regions in our annotation in a strand-specific manner. These regions were then overlapped

with the putative recursive junctions using overlapsAny from IRanges (Lawrence et al, 2013) such that those with an RS site acting as a 5′ splice site were overlapped with the 5′ proximal regions and likewise those with an RS site acting as a 3′ splice site were overlapped with the 3′ proximal region.

### Branchpoint distance from the 3′ splice site

The distance between the branchpoint and the 3′ splice site, calculated as the absolute value of the position of the 3′ splice site minus the position of the branchpoint (i.e. the length of the tail not including the branchpoint), was determined for mapped sequences. Those with distances outside the range of 10–60 nt were discarded, based on the distribution of this distance in previous work (Gao et al, 2008; Mercer et al, 2015; Pineda & Bradley, 2018) and a manual review of our initial results using the Mercer et al (2015) sequencing data as a test set.

### Filtering UpSet plot

The results (the number of lariat reads, the number of unique lariat reads, the number of unique lariat reads with one mismatch, with at least one mismatch, and the calculated FDR for each read type) from each of the filtering steps were collected manually for each permutation in order to optimize the order and to determine the essentiality of each set. These results were then manually formatted in Excel into a table compatible with the R package ComplexUpset (Lex et al, 2014; Krassowski, 2020), with which they were then plotted.

### Modeling informative reads

For each constitutive intron in the human genome, 1,000 positions in the portion of the intron that makes up the lariat loop (i.e., from the start of the intron to the estimated branchpoint position) were chosen at random with replacement. The median U2 branchpoint position of 25 nucleotides upstream of the 3′ splice site (Taggart et al, 2017) was assumed. A position was considered able to produce an informative lariat read if it was (1) at least 20 nucleotides upstream of the estimated branchpoint position (minimum branchpoint-mapping region) and (2) no less than n nucleotides away from the estimated branchpoint plus 20 (minimum 5′-mapping region), where n is a given sequencing read length. Read lengths of 75, 100, and 150 nucleotides were tested with this model to evaluate common sequencing read lengths.

The fraction of 1,000 random positions for each intron that fulfilled the above criteria was calculated for each read length. The probabilities for 100 nucleotide reads were then used to generate the expected distribution of lariat sizes from constitutive introns assuming no differences in gene expression. Of the non-linear decay functions evaluated, the model that fit the data the best was $y = ax^b$, where $x$ is the length of the intron.

### phastCons

Conservation scores for vertebrates (phastCons100way.UCSC.hg19), placental mammals (phastCons46wayPlacental.UCSC.hg19), and primates (phastCons46wayPrimates.UCSC.hg19) were gathered using GenomicScores in Bioconductor (version 3.14). Base levels of conservation were determined by randomly selecting one of the unfiltered, randomly chosen decoy sites for each intron considered in our analysis. The confidence intervals represent one SD, which was determined by bootstrapping (n = 100) the mean.

### External site analysis

Previous work that identified recursive splice sites in human introns genome-wide using sawtooth sequencing or lariat sequencing was queried for tables of RS sites or RS junctions from which RS sites could be derived. The works that met the criteria included the following: Duff et al (2015); Sibley et al (2015); Zhang et al (2018); and Wan et al (2021).

- Duff et al (2015): RS sites were acquired directly from the table located at https://static-content.springer.com/esm/art%3A10.1038%2Fnature14475/MediaObjects/41586_2015_BFnature14475_MOESM124_ESM.xlsx
- Sibley et al (2015): sites were parsed from the table located at https://www.ncbi.nlm.nih.gov/pmc/articles/PMC4471124/bin/NIHMS62941-supplement-Table_S1.xlsx

   For the short-read sites, the RS site was inferred from the junction ID and the annotation of the "first part" and "second part" coordinate and type (whether the first "part" belonged to the exon or intron side) columns using the genomic sequence annotations in the table to manually check a subset of sequences; then, they were filtered according to the table classification as "recursive." This column appears to be largely determined by adherence to the article's described requirement of >11 pyrimidines, a suitable 3′ss motif (annotated), and a suitable 5′ss motif (annotated); however, we were not able to reproduce the exact count reported in the article, so we defaulted to the table's classification as there may have been an additional factor we overlooked. The junction IDs are shared between these sites and the linear regression sites used for the sawtooth analysis. Thus, we were able to use the intron information provided with the linear regression table to annotate a subset of the short-read sites.
   For the sawtooth linear regression analysis sites, the RS site annotations were populated by joining the table with the shared junction IDs. Because of the short-read annotations containing only exon–intron or intron–exon annotations, there are no RS–RS sites, which may plausibly exist in the linear regression annotation, so this may be a slight undercount. We attempted to annotate the RS sites based on a comparison of the junction coordinates with the intron coordinates, but because of inconsistencies of unknown origin with the strand annotation relative to the short-read sites, we were unable to use this method, which would have allowed for the identification of such sites. We defined sawtooth RS sites as those that were annotated with "double significance and improved gradient" following the article's methods.
- Wan et al (2021): RS sites were derived from the annotations of RS junctions acquired by direct correspondence with the authors. The site locations were calculated using the start, end, and strand coordinates and the annotation of which end(s) of the junctions the RS site(s) were located on.

- Zhang et al (2018): RS sites were acquired directly from the tables located at https://doi.org/10.1371/journal.pgen.1007579.s011 (XLSX); https://doi.org/10.1371/journal.pgen.1007579.s012 (XLSX); and https://doi.org/10.1371/journal.pgen.1007579.s013 (XLSX). We used the site selection criteria described in the Materials and Methods section (a fold change greater than 2 and a P-value less than 0.01) to filter for sites that met it in at least one of the samples. We did not consider the manual review criteria as it was not parsable from the table's text and was described as optional in the article's methods.

The counts in the table were generated such that they included all provided sites for the given methods that were not supported by a "higher" evidence level. A subset of sites were annotated as belonging to multiple introns; therefore, we weighted based on how many introns they were said to belong to and took the median intron length using the *weighted.median* function from the R package limma (Ritchie et al, 2015). They were then also assessed for the presence of the given motifs at the RS site.

### Intersections of our sites with external sites

For the Euler sites and UpSet plots, we included those sites that were annotated as recursive by the authors. The Euler plot was generated using the R package eulerr (Larsson 2021). Sites were intersected using just the chromosome and site information without regard to the strand given the uncertainty about the strand annotations in a subset of the Sibley sites. The UpSet plot was generated using the R package UpSetR (Gehlenborg 2019).

## Distal exonic lariat analysis

### Lariat detection

The same mapping (but not filtering) steps were followed as above, except that internal cassette exons and their upstream introns were used to build all possible junctions of paired annotated 5′ splice sites and potential distal exonic 3′ splice sites, regardless of sequence. Two filters were used: the same branchpoint distance filter applied to the intronic lariats (10–60 nt) and one that required the branchpoint position to be within the exon coordinates (and not the start or end base). Exons wre considered internal if they were not first or last across all RefSeq transcripts that included them (across all gene names) in the RefSeq annotation.

### Exon inclusion analysis

Exon inclusion was estimated across the 16 tissues in the BodyMap 2.0 database as described previously (Dvinge et al, 2014). A Kolmogorov–Smirnov test was performed using the function ks. test (stats R package, version 4.3.0) with a null hypothesis of "not greater" for the median percent spliced in (psi) across all BodyMap tissues comparing internal cassette exons where the last three nucleotides were YAG with all other internal cassette exons.

### Maximum entropy scores

MaxEntScan scores as defined in Yeo and Burge (2004) were calculated using an R function that called the Perl scripts available for download from http://hollywood.mit.edu/burgelab/maxent/download/ for the 3′ splice sites. Traditionally, MaxEntScan scores for conventional 3′ splice sites require the 20 bases of the intron and 3 bases of the exon; here, for calculating the score for distal exonic RS sites, we used the 20 bases upstream of and including the last base of the putative distal RS-exon and the three bases of the downstream intron.

## Sequence logo plots

Sequence logo plots were generated using an adapted version of the seqLogo R package (Bembom & Ivanek, 2022).

## Exon age categorization

Exon age annotations were obtained from Corvelo and Eyras (2008) at https://static-content.springer.com/esm/art%3A10.1186%2Fgb-2008-9-9-r141/MediaObjects/13059_2008_2003_MOESM2_ESM.zip, which categorized exons using the conservation of sequences 20 nt upstream of and 20 nt downstream of the 5′ and 3′ splice sites of annotated exons. They used a comparison of five species (human, mouse, cow, chicken, and *Tetraodon*), so primate-specific exons are those that were in humans, but not in mouse or cow (and therefore may be human-specific or primate-specific); mammalian-specific exons were those in humans, mice, and cows. Vertebrate and older exons were those observed in all five species. The reported hg18 coordinates were converted to hg19 using the liftOver utility provided by the UCSC Genome Browser (Meyer et al, 2013) using default settings (minimum ratio of bases that must remap = 0.95; allow multiple output regions; minimum hit size = 0; minimum chain size = 0). We performed the binomial exact test ($\alpha$ = 0.05, alternative = "greater") using the *binomial.test* function from the R package stats (R Core Team, 2022).

## Plots

All plots, unless otherwise specified, were generated in R using ggplot2 (Wickham, 2016).

## Tables

Tables included in the text were generated using the R package gt (Iannone et al, 2022).

# Data Availability

All datasets used in this study are publicly available or previously published. A table of accession information for the samples used can be found in the Supplementary Information (Table S1). The code used in this article relies on previously published software (see the Materials and Methods section above for details and citations).

# Supplementary Information

## Acknowledgements

We thank current and former members of the Bradley laboratory for helpful discussions, in particular, Guo-Liang Chew and Jose Pineda, and the Fred Hutch Scientific Computing Core for their support. We also thank Jose Pineda for his helpful review of the article. RK Bradley was supported in part by the National Institutes of Health (NIH)/National Heart, Lung, and Blood Institute (NHLBI) (R01 HL151651); NIH/National Cancer Institute (NCI) (R01 CA251138); NIH/NHLBI (R01 HL128239); Blood Cancer Discoveries Grant Program through the Leukemia & Lymphoma Society, Mark Foundation for Cancer Research, and Paul G Allen Frontiers Group (8023-20); and Department of Defense Breast Cancer Research Program (W81XWH-20-1-0596). RK Bradley is a Scholar of The Leukemia & Lymphoma Society (1344-18) and holds the McIlwain Family Endowed Chair in Data Science. This research was supported in part by the NIH/NCI (Cancer Center Support Grant P30 CA015704). DB Udy was supported in part by NIH/NIGMS (T32 GM007270).

### Author Contributions

ER Hoppe: conceptualization, investigation, visualization, and writing—original draft, review, and editing.
DB Udy: visualization and writing—original draft, review, and editing.
RK Bradley: conceptualization and writing—original draft, review, and editing.

### Conflict of Interest Statement

The authors declare that they have no conflict of interest.

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
