## [Reviewer comments · Life Science Alliance]

Life Science Alliance

Recursive splicing discovery using lariats in total RNA sequencing

Emma Hoppe, Dylan Udy, and Robert Bradley

DOI: <https://doi.org/10.26508/lsa.202201889>

Corresponding author(s): Robert Bradley, Fred Hutchinson Cancer Center

Review Timeline:

Submission Date:	2022-12-23
Editorial Decision:	2023-02-02
Revision Received:	2023-04-14
Editorial Decision:	2023-04-19
Revision Received:	2023-04-20
Accepted:	2023-04-21

Transaction Report:

February 2, 2023

Re: Life Science Alliance manuscript #LSA-2022-01889

Dr. Robert K. Bradley
Fred Hutchinson Cancer Center
Computational Biology Program
FHCRC
1100 Fairview Avenue North
Seattle, WA 98109

Dear Dr. Bradley,

Thank you for submitting your manuscript entitled "Recursive splicing discovery using lariats in total RNA sequencing" to Life Science Alliance. The manuscript was assessed by expert reviewers, whose comments are appended to this letter. We invite you to submit a revised manuscript addressing the Reviewer comments.

Thank you for this interesting contribution to Life Science Alliance. We are looking forward to receiving your revised manuscript.

Sincerely,

B. MANUSCRIPT ORGANIZATION AND FORMATTING:

Reviewer #1 (Comments to the Authors (Required)):

Here the authors use computational approach to identify recursive splicing sites within introns. They used total RNA sequencing data and identify a number of new recursive splice sites not previously known. They show that long introns are not necessarily a requirement for recursive splicing, as previously published. They provide evidence that all sizes of introns, including small ones can use this kind of specialized splicing event. They find that while choice of such splice sites is stochastic, some may be conserved and regulated. The authors also provide examples where recursive splicing is used for exon exclusion. Overall, this clear and well-written manuscript provides support for more a broader use of recursive splicing in the human transcriptome. I support its publication.

Reviewer #2 (Comments to the Authors (Required)):

This study by Hoppe et al. aims to expand our knowledge of recursive splicing in human cells by using unbiased methods to identify and characterize sites of recursive splicing. To do so, they search for high-throughput sequencing reads that support the presence of intron lariats from recursive splicing. Using these sites, they are able to show a broader range of introns with recursive splicing, higher than expected conservation across vertebrates, and roles for these sites in regulating alternative splicing. Overall, this study is well written and presents solid analyses in support of an expanded role for recursive splicing in mammalian cells. However, there a few analysis details and points that I think need to be clarified before publication.

Specific Comments:

- (1) I could not find a place where the authors detail the samples used for these analyses. In the methods, they say that they search the SRA to find human nascent RNA-seq datasets but give no context for which samples were ultimately chosen. I understand that they likely used a lot of samples, but think that there needs to be a mention somewhere in the Results or Methods section about the actual samples used to give context for how to compare their results with other studies. The authors similarly do not mention how many datasets or total reads were used, which provides important context for their power to identify recursive sites relative to other studies.
- (2) On a related note, could the lack of overlap (Figure 1C) between the sites identified here and other studies be due to differences in cell types? The authors also do not show overlap with the Wan et al. 2021 paper, which identified recursive sites and uses nascent RNA-seq data.
- (3) Can the authors comment at all on the probably that these recursive sites are being used consistently to aid in splicing of recursive introns or stochastically, such that any given recursive site is being used a minority of the time at random (as suggested by Wan et al. 2021)? This might be a useful discussion in the context of the suggestion that these recursive sites in shorter introns are aiding in the choice of alternative splice sites for those introns.

REVIEWER 1

Here the authors use computational approach to identify recursive splicing sites within introns. They used total RNA sequencing data and identify a number of new recursive splice sites not previously known. They show that long introns are not necessarily a requirement for recursive splicing, as previously published. They provide evidence that all sizes of introns, including small ones can use this kind of specialized splicing event. They find that while choice of such splice sites is stochastic, some may be conserved and regulated. The authors also provide examples where recursive splicing is used for exon exclusion. Overall, this clear and well-written manuscript provides support for more a broader use of recursive splicing in the human transcriptome. I support its publication.

We thank the reviewer for the kind and encouraging comments on our manuscript.

REVIEWER 2

This study by Hoppe et al. aims to expand our knowledge of recursive splicing in human cells by using unbiased methods to identify and characterize sites of recursive splicing. To do so, they search for high-throughput sequencing reads that support the presence of intron lariats from recursive splicing. Using these sites, they are able to show a broader range of introns with recursive splicing, higher than expected conservation across vertebrates, and roles for these sites in regulating alternative splicing. Overall, this study is well written and presents solid analyses in support of an expanded role for recursive splicing in mammalian cells. However, there are a few analysis details and points that I think need to be clarified before publication.

We thank the reviewer for their appreciation of our study's contributions to the literature on recursive splicing.

Specific Comments:

I could not find a place where the authors detail the samples used for these analyses. In the methods, they say that they search the SRA to find human nascent RNA-seq datasets but give no context for which samples were ultimately chosen. I understand that they likely used a lot of samples, but think that there needs to be a mention somewhere in the Results or Methods section about the actual samples used to give context for how to compare their results with other studies. The authors similarly do not mention how many datasets or total reads were used, which provides important context for their power to identify recursive sites relative to other studies.

We thank the reviewer for pointing out this oversight. We edited **Table S1** so that it clearly states the relevant details and statistics for each sample that we analyzed in our study and additionally states the total number of reads that were included in our analysis over all samples. We additionally state these important statistics in the revised text. We edited the Methods ("Dataset selection and annotation") and Data Availability sections to clearly state that the relevant details and statistics for the samples that we analyzed are found in **Table S1**.

On a related note, could the lack of overlap (Figure 1C) between the sites identified here and other studies be due to differences in cell types?

We thank the reviewer for highlighting this important point. As the reviewer notes, the impact of cell type differences is an intriguing and important question to highlight that could contribute to the relatively low overlap that we observed. We edited the manuscript to emphasize this important point:

The limited overlap that we observed could additionally arise in part from differences in the cell types that were analyzed in each study. Our analysis utilized a diverse selection of cell lines and patient-derived fibroblasts (**Table S1**) which were distinct from the cell types analyzed in other works (Duff et al. (2015): selected adult tissue samples; Sibley et al. (2015): adult brain tissue; Zhang et al. (2018): ovarian, embryonic stem cell, and embryonic stem cell-derived forebrain cell lines; Wan et al. (2021): bronchial epithelial cell line).

The authors also do not show overlap with the Wan et al. 2021 paper, which identified recursive sites and uses nascent RNA-seq data.

We thank the reviewer for this excellent suggestion. We revised the manuscript to include the recursive sites identified by Wan et al, 2021 in our analysis of overlap between different studies of recursive splicing. This new analysis is found in the revised **Table 1** and **Figure S2**.

Comment 3: *Can the authors comment at all on the probability that these recursive sites are being used consistently to aid in splicing of recursive introns or stochastically, such that any given recursive site is being used a minority of the time at random (as suggested by Wan et al. 2021)? This might be a useful discussion in the context of the suggestion that these recursive sites in shorter introns are aiding in the choice of alternative splice sites for those introns.*

We thank the reviewer for raising this interesting point. To tackle this question, we mapped lariats for conventional splicing across the genome and compared conventional versus recursive splicing rates in the introns for which recursive splicing had been identified. Although there was not enough data to definitively assess whether each intron appears to be regulated or stochastic (and this may depend on factors that the reviewer mentioned above, like cell type), we sought to identify whether there was a difference between shorter and longer introns in the usage of recursive versus conventional splicing. As suggested by Wan et al, 2021 and highlighted by the reviewer, this analysis supported a preference for conventional splicing among shorter introns, with a relatively strong association in the expected direction. We describe this new analysis in the main text in the section “Lariat sequencing identifies recursive splicing in diverse introns”:

Finally, we compared the relative rates of identification of conventional and recursive lariats for the introns in which we identified recursive splicing to determine whether recursive splicing appeared to be occurring infrequently or stochastically (as suggested by Wan et al., 2021) versus as a more common or dominant mode of splicing. Approximately half of the introns (46 of 100) with recursive lariats had at least one conventional lariat that we identified. For short introns (<1,000 nt), we observed a preference for lariats arising from conventional versus recursive splicing (**Fig S4A and S4B**). A Spearman’s rank correlation performed on the relative number of unique recursive lariats with mismatches compared to the intron length in those introns with at least one lariat of each type revealed a positive association between intron length and the relative number of recursive lariats ($\rho = 0.4912$, $p = 0.0005272$). These data suggest that certain introns identified here, especially smaller introns, may be primarily subject to recursive splicing as a stochastic and less frequent mode of splicing, while others show a preference for recursive over conventional splicing.

We also highlighted this point in the revised Discussion:

Our work here indicates that the frequency of use of recursive versus conventional splicing may be a function of intron length, with stochasticity playing a stronger role in smaller introns. However, further work will be needed to confirm this observation and elucidate the role, if any, recursive splicing plays in regulating alternative introns, as the work here primarily focuses on constitutive introns.

April 19, 2023

RE: Life Science Alliance Manuscript #LSA-2022-01889R

Dr. Robert K. Bradley
Fred Hutchinson Cancer Center
Computational Biology Program
FHCRC
1100 Fairview Avenue North
Seattle, WA 98109

Dear Dr. Bradley,

Thank you for submitting your revised manuscript entitled "Recursive splicing discovery using lariats in total RNA sequencing". We would be happy to publish your paper in Life Science Alliance pending final revisions necessary to meet our formatting guidelines.

- please upload your manuscript text as an editable doc file
- please upload your main and supplementary figures as single files and add a separate figure legend section to the main manuscript text
- please upload your table files as excel or doc files or make sure that they're included in the doc file of your main manuscript text
- please add the Twitter handle of your host institute/organization as well as your own or/and one of the authors in our system
- please use the [10 author names, et al.] format in your references (i.e. limit the author names to the first 10)

A. FINAL FILES:

B. MANUSCRIPT ORGANIZATION AND FORMATTING:

Thank you for your attention to these final processing requirements. Please revise and format the manuscript and upload materials within 3 days.

Sincerely,

April 21, 2023

RE: Life Science Alliance Manuscript #LSA-2022-01889RR

Dr. Robert K. Bradley
Fred Hutchinson Cancer Center
Computational Biology Program
FHCRC
1100 Fairview Avenue North
Seattle, WA 98109

Dear Dr. Bradley,

Thank you for submitting your Research Article entitled "Recursive splicing discovery using lariats in total RNA sequencing". It is a pleasure to let you know that your manuscript is now accepted for publication in Life Science Alliance. Congratulations on this interesting work.

DISTRIBUTION OF MATERIALS:

Again, congratulations on a very nice paper. I hope you found the review process to be constructive and are pleased with how the manuscript was handled editorially. We look forward to future exciting submissions from your lab.

Sincerely,
